# Designing DCAT-AP extensions for common European data spaces: The EHDS HealthDCAT-AP Case Study

Pascal Derycke[1*], Beatriz J. Barros[1], Nienke M. Schutte[1], Charles-Andrew Vande Catsyne[1] and Martina Bargeman Fonseca[1]

[1] Sciensano, Rue Juliette Wytsmanstraat 14, 1050 Brussels, Belgium

### Abstract

The European Health Data Space (EHDS) Regulation aims to enable efficient cross-border discovery, access, and secure reuse of electronic health data (i.e. sensitive and non-sensitive datasets) for research, innovation, and policymaking. To meet EHDS regulatory requirements, Health Data Access Bodies (HDABs) are obliged to maintain an interconnected public national dataset catalogue and health data holders must provide standardised descriptions of datasets, with the aim to enable efficient discovery and interoperability across Europe. While the DCAT Application Profile (DCAT-AP) is widely adopted for general data cataloguing across the EU due to its robustness and simplicity, its generic scope lacks essential health-specific metadata elements which are required to assist the implementation of the EHDS, such as data sensitivity levels, access regulatory procedures, health-specific terminologies, and quality annotations.

This paper presents the EHDS-related extension of the DCAT Application profile (HealthDCAT-AP) case study, detailing how a domain-specific extension of DCAT-AP was designed to address these gaps. It outlines the methodological approach involving use-case analysis, careful requirements definition, and stakeholder engagement, which resulted in a concise and interoperable fit for purpose metadata extension. The resulting HealthDCAT-AP, validated through a multi-country pilot implementation, followed by a refinement and validation phase and a pan-European public consultation, demonstrates a practical blueprint for other specialised DCAT-AP profiles across domain-specific European data spaces.

### Keywords

Metadata, DCAT-AP extensions, common EU data spaces, interoperable dataset catalogues, European Health Data Space (EHDS), HealthDCAT-AP

## 1. Introduction

The European Health Data Space (EHDS) Regulation [1] addresses the longstanding fragmentation in the use and accessibility of electronic health data across the EU. This inserts itself in the wider EU Digital Strategy, which foresees the creation of common European data spaces[1] that promote easier access and reuse of data for several strategic fields (including health, agriculture, energy), in a trustworthy and secure way. The aim is also that such data spaces become gradually interconnected, to better ensure that data can create impact in solving global and multisectoral challenges. In this context, the EHDS supports the alignment of practices and the promotion of interoperability to simplify data-sharing, foster collaboration among Member States, and encourage the efficient use of resources. Common specifications such as the Data Catalog Vocabulary – Application Profile (DCAT-AP) [2] play a vital role by providing a consistent framework that ensures coherence between the EHDS and other European data spaces.

The EHDS, dedicated to the health domain, introduces new obligations for Member States and health data holders regarding the secondary use of electronic health data. Central to these obligations is the creation of standardised, machine-readable descriptions of datasets to facilitate efficient discovery, access, and reuse across borders. The widely adopted DCAT-AP provides a robust

---

[*] Corresponding author.

✉ pascal.derycke@sciensano.be (P. Derycke); beatriz.barros@sciensano.be (B. Barros); nienke.schutte@sciensano.be (N. Schutte); Charles-Andrew.VandeCatsyne@sciensano.be (C-A. Vande Catsyne); martina.fonseca@sciensano.be (M. Bargeman Fonseca)

Ⅾ 0000-0002-4658-5185 (P. Derycke); 0000-0002-2593-0824 (B. Barros); 0000-0002-8064-2569 (N. Schutte); 0000-0001-6283-9029 (C-A. Vande Catsyne); 0000-0003-3020-8112 (M. Bargeman Fonseca)

[1] Shaping Europe's digital future: Common European Data spaces

foundation for general data cataloguing within the EU; however, it lacks specific metadata elements required to assist the implementation of the EHDS, including sensitivity classifications, permitted usage purposes, health coding references, and structured quality indicators.

To address these gaps, the HealthData@EU pilot project[2] developed HealthDCAT-AP[3], a dedicated, EHDS-specific extension of the DCAT-AP that is based on the Data Catalogue Vocabulary (DCAT)[4] developed by W3C. This paper describes the systematic design of HealthDCAT-AP, highlighting key methodological steps, such as stakeholder involvement through working groups and surveys, detailed requirement identification based on practical use cases and EU regulatory requirements, and careful adherence to existing linked data and interoperability principles. The resulting HealthDCAT-AP introduces a minimal yet comprehensive set of extensions—including new classes, metadata properties, essential controlled vocabularies, usage notes, and refined cardinalities. The profile remains fully compatible with DCAT-AP, ensuring seamless integration with existing EU dataset catalogues[5]. A validation phase, conducted in a multi-country sandbox environment using real-world examples and supported by the development of the HealthData@EU Central Platform[6], demonstrated the profile's practical utility and its capacity to enhance discoverability, interoperability, and legal compliance across the HealthData@EU infrastructure.

In this context of interconnected European data spaces envisioned by the EU Digital Strategy[7], the development of HealthDCAT-AP provides a robust reference model for future domain-specific DCAT-AP extensions. This highlights the importance of the approach presented in this paper, which supports precedent efforts for other sectoral extensions to ensure harmonisation and interoperability within the wider European data ecosystem. Key technical outcomes include the implementation of a dedicated metadata editor to support health data holders, the publication of the extension specification as a single source of truth for catalogue implementers, and the release of a literacy platform with tooling for onboarding and adoption[8]. The profile is supported by the HealthData@EU Central Platform, and all materials are openly licensed under CC BY 4.0, facilitating reuse by HDABs. A FAIR[9] Data Point-based open-source cataloguing solution enables integration by health data publishers, while Shapes Constraint Language (SHACL)[10] validation tooling ensures syntactic and semantic conformance. The importance of a well-structured, inclusive stakeholder process serves also as a valuable reference for future domain-specific DCAT-AP extensions. Together, these insights demonstrate that targeted yet minimal extensions can deliver significant improvements in data usability, interoperability, and cross-border collaboration, providing a practical blueprint for other EU data spaces yet to be established.

## 2. Introduction – Policy context & problem statement: why HealthDCAT-AP?

### 2.1. Regulatory driver: The European Health Data Space (EHDS)

The EHDS Regulation adopted in March 2025 [1] represents a critical component of Europe's data strategy[11], aiming to address significant fragmentation in electronic health data access and use across EU Member States. Under this EU regulatory framework, Member States are required to establish dedicated HDABs that facilitate and govern secondary uses of health data for research, innovation and policy-making. Article 77 of the EHDS explicitly requires HDABs to maintain interoperable public national dataset catalogues. These catalogues must contain standardised, machine-readable

---

[2] Grant agreement no 101079839
[3] Draft HealthDCAT-AP specification
[4] Data Catalog Vocabulary (DCAT)
[5] data.europe.eu - Official portal for European data
[6] HealthDataEU Central Platform | Open-source components
[7] https://digital-strategy.ec.europa.eu/en
[8] HealthDCAT-AP | Health Information Portal
[9] Platform that exposes metadata for machines according to the FAIR data principles (Findable, Accessible, Interoperable, Reusable) https://specs.fairdatapoint.org/fdp-specs-v1.2.html
[10] Shapes Constraint Language (SHACL)
[11] A European strategy for data | Shaping Europe's digital future

metadata records, which are to be aggregated into a unified EU-level dataset catalogue. The EHDS, through this harmonised metadata framework, strives to create a secure, interoperable and, federated environment - the HealthData@EU infrastructure - in which electronic health datasets are discoverable, accessible, and reusable across borders, thus unlocking the considerable potential of European health data resources.

A cornerstone of the EHDS Regulation is the obligation placed upon health data holders to ensure datasets under their control are described in a standardised manner. These descriptions are to be integrated into federated standardised machine-readable dataset catalogues at the national and EU levels. Such standardisation is essential for facilitating data discovery and enhancing interoperability, directly supporting cross-border data use.

**Table 1**
Reference EHDS Articles

---

**EHDS Article 60 – 3**: "*The health data holder shall communicate to the health data access body a description of the dataset it holds in accordance with Article 77. The health data holder shall, at a minimum on an annual basis, check that its dataset description in the national dataset catalogue is accurate and up to date.*"

**EHDS Article 60 – 5** states that non-personal electronic health data must be exposed through "*trusted open databases*", implying at least one openly accessible distribution per dataset.

**EHDS Article 77 – 1** requires HDABs to maintain national "*publicly available and standardised machine-readable dataset catalogues*" whose records are harvested into a single EU catalogue and, by extension, into the National Single Information Points mandated by the Data Governance Act (DGA) [3].

---

## 2.2. Problem statement: The need for a health-specific metadata standard

While DCAT-AP has been broadly adopted by EU institutions and Member States to facilitate data sharing across various sectors, its generic nature limits its effectiveness for health data in scope of the EHDS. Health datasets inherently include sensitive and highly regulated data elements, often collected under strict ethical and legal frameworks such as the General Data Protection Regulation (GDPR) [4] and specific national health data regulations. As such, standard metadata elements available in DCAT-AP alone fall short of meeting crucial EHDS-specific requirements. Health datasets necessitate additional metadata capabilities, including explicit representation of data sensitivity (open, protected, sensitive), access regulatory procedures, permitted usage purposes (e.g., research, public health surveillance, innovation), dataset structure (e.g., data dictionary), precise linkage to medical terminologies (e.g., ICD-10, SNOMED CT), main characteristics of the dataset (e.g. population coverage, age) and structured data quality indicators mandated explicitly by EHDS Articles 77–80.

Without these enhancements, discoverability remains limited, since health data users cannot filter by health relevant facets (e.g. age range, disease). This also restricts accessibility and, consequently, the appropriate secondary use of data. As a result, compliance with the EHDS regulatory requirements remains uncovered.

## 2.3. The regulatory path towards HealthDCAT-AP

Recognising the need for a common metadata model explicitly aligned with the EHDS regulatory framework, the European Commission launched the HealthData@EU pilot project to develop and test a dedicated health extension of DCAT-AP. Through systematic stakeholder engagement - including targeted workshops and iterative surveys involving representatives from HDABs, health data holders, metadata or linked data experts, researchers, and policymakers - a structured and minimal set of metadata enhancements was defined. This process ensured that each new metadata term was rigorously justified, broadly supported, and closely aligned with both regulatory and practical requirements. Building on the outcomes of the pilot, the TEHDAS2 Joint

Action[12], involving 29 European countries, further prepared the ground for harmonised implementation. Through a series of technical workshops (for experts) and a pan-european public consultation, TEHDAS2 contributed to aligning Member State perspectives and validating the approach. The EHDS HealthDCAT-AP is now positioned to serve as the foundation for the EHDS implementing acts (planned for 2027), embedding the extension into a legal and operational framework for metadata interoperability across Europe.

## 3. Background: Choosing and extending DCAT-AP (DCAT-AP family, extending without breaking)

### 3.1. What is DCAT-AP?

DCAT-AP is an RDF-based metadata standard developed to facilitate the interoperability and exchange of metadata across different dataset catalogues and domains in the European Union. Built upon the W3C DCAT vocabulary, DCAT-AP provides a structured set of classes and properties for describing datasets and data services. Its core classes, such as *dcat:Catalog*, *dcat:Dataset*, and *dcat:Distribution*, are designed to support the consistent description, discovery, access, and reuse of datasets. DCAT-AP ensures that metadata are exposed in both machine-readable and human-readable formats, enabling automated processing as well as user-friendly interfaces. Its structured format enhances dataset searchability, simplifies data integration, and promotes cross-border data sharing and reuse, in alignment with the FAIR data principles [5]. DCAT-AP is widely adopted by EU Member States and institutions, providing a standardised approach for metadata publication across Europe.

### 3.2. What DCAT-AP is not

While DCAT-AP effectively facilitates data exchange across data portals, it is not intended as a comprehensive internal data management framework nor to replace existing established domain-specific metadata models:

- DCAT-AP's design, which leverages RDF (Resource Description Framework)[13], typically differs significantly from internal organisational data storage systems, often based on relational databases or tabular structures. Consequently, DCAT-AP does not directly replace or manage operational data systems; instead, it acts primarily as a standard for external data cataloguing, interoperability, and discoverability. Moreover, DCAT-AP alone does not inherently ensure data quality or semantic accuracy, nor describe the internal structure of datasets specific to domains like health, mobility, geospatial or statistical data.
- HealthDCAT-AP is not intended to replace existing domain-specific metadata models or other data governance practices used in the health sector. Instead, it acts as a common framework - a metadata lingua franca - that enables interoperability across diverse systems and metadata standards. HealthDCAT-AP is designed to complement and coexist with standards and formats applied in the health domain such as HL7 FHIR[14], CDISC[15], and ISO/IEC 11179[16], supporting cross-system data exchange while enabling a unified approach to metadata description across the EHDS.

### 3.3. Why DCAT?

DCAT provides a well-established, flexible, and widely adopted standard for describing datasets and data catalogues. It is recognised internationally for its robustness and simplicity, allowing straightforward integration and compatibility with existing dataset cataloguing practices. By using a

---

[12] Second Joint Action Towards the European Health Data Space – TEHDAS2 (Grant agreement no 101176773)
[13] Resource Description Framework
[14] HL7 FHIR Specification
[15] CDISC | Clear Data. Clear Impact.
[16] ISO/IEC 11179-1:2023 - Information technology

common vocabulary, DCAT acts as a lingua franca for dataset metadata - ensuring that descriptions are consistently structured, machine-readable[17], machine-actionable [18]and easily understood across diverse platforms and institutions. Its core design supports both interoperability and extensibility, allowing domains such as health to tailor the model with additional metadata elements without sacrificing interoperability.

## 3.4.    State of play of EU dataset catalogues

The European Commission (EC) has actively promoted DCAT-AP's adoption as the standard metadata framework for dataset catalogues across Member States, underpinning Europe's open-data strategy and broader digital transformation goals. Most EU Member States already use DCAT-AP within their national dataset catalogues. The portal for European data (data.europa.eu) aggregates almost two million datasets, including about 27000 health-related records. Growth has been propelled by successive legislation – Open Data Directive [6], INSPIRE Directive [7], Data Governance Act [3] or the High Value Dataset Implementing Regulation [8] – which mandate machine-readable metadata and single information points[19] in every Member State. Health dataset catalogues required by the EHDS must federate into this ecosystem, making reuse of the established DCAT-AP infrastructure the most cost-effective strategy. The ongoing use of DCAT-AP aligns with the EC's overarching data policy objectives, including transparency, interoperability, and the establishment of integrated Common European Data Spaces, such as the EHDS.

## 3.5.    Why create an application profile?

While DCAT-AP provides a robust general-purpose foundation, specific domains - such as geospatial, statistical, or health data - often require additional, targeted metadata elements and controlled vocabularies. Creating a domain-specific Application Profile (AP) allows the precise tailoring of the DCAT model to address these specialised needs. For example, StatDCAT-AP[20] adds statistical dimensions, while GeoDCAT-AP[21] incorporates specific elements for geospatial data, enabling the mapping of metadata from the INSPIRE standard (expressed in XML using ISO 19115) to DCAT-AP.  For datasets subject to the requirements imposed by the High-Value Dataset Implementing Regulation (focussed on strategic datasets of consequence to 6 thematic categories: geospatial, earth observation and environment, meteorological, statistics, companies and company ownership, and mobility), the dedicated DCAT-AP HVD profile[22] is also promoted.

These domain-specific application profiles maintain compatibility with core DCAT-AP while providing domain experts with the precision needed to accurately describe complex datasets, thus significantly improving dataset discoverability, usability, and interoperability within their respective data communities. For health data, this is indispensable: health datasets must convey key attributes such as population coverage, legal bases for processing, health-specific coding standards, and data quality metrics - elements that fall outside the scope of generic DCAT-AP.

## 3.6.    How to extend DCAT-AP without breaking it?

Extending DCAT-AP must be done carefully to avoid compromising the interoperability benefits already achieved. The guiding principle for DCAT-AP extensions is minimalism and compatibility: new terms should only be introduced when existing DCAT-AP elements or existing domain vocabularies cannot sufficiently describe the datasets. Such extensions must respect namespace isolation, meaning new terms should be defined within dedicated namespaces clearly distinct from core DCAT-AP terms. Extensions must also avoid redefining or changing the semantics of existing DCAT-AP terms, ensuring any dataset description remains valid according to the original

---

[17] Information structured in a format that can be automatically read and processed by a computer
[18] Information with structure and semantics to enable automated decision-making or operations
[19] European Single Access Point: Harvesting guidelines for Member States
[20] StatDCAT Application Profile for data portals in Europe | Interoperable Europe Portal
[21] GeoDCAT Application Profile for data portals in Europe | Interoperable Europe Portal
[22] DCAT Application Profile for High-Value Datasets | Interoperable Europe Portal

standard. Finally, rigorous validation procedures, often leveraging SHACL, are essential to ensure that metadata records conform consistently to both the original DCAT-AP specification and the newly introduced domain-specific enhancements.

Extension rules promoted by SEMIC[23]:

1. **Respect core conformance** - all DCAT-AP mandatory properties remain mandatory.
2. **Avoid overlap** - new classes or properties must not duplicate existing ones.
3. **Keep vocabularies aligned** - reuse the EU controlled lists already mandatory in DCAT-AP, adding health-specific lists only when justified.
4. **Adjust cardinalities prudently** - the requirement levels (Mandatory, Recommended, Optional) may be upgraded to mandatory or recommended, but never downgraded.

Because DCAT-AP records are RDF graphs, extra triples can be added without disrupting existing APIs or harvesters, ensuring that legacy catalogues continue to operate while gradually exposing richer health metadata. These principles were rigorously applied in designing HealthDCAT-AP, ensuring that the resulting profile remains lightweight, focused, and fully interoperable with existing European metadata infrastructures.

## 4. Methodology: roadmap and stakeholder engagement

### 4.1. HealthDCAT-AP Design Roadmap

The development of HealthDCAT-AP followed a structured, iterative evidence-driven process designed to ensure careful alignment with both user needs and the EHDS regulatory requirements. The roadmap was organised into several sequential yet overlapping phases, each dedicated to achieving clear intermediate outcomes:

**Phase 1: Identification of requirements**
In this initial phase, a detailed assessment was conducted to gather and categorise the specific requirements derived from the EHDS Regulation and other horizontal EU data policies, the FAIR data principles, and existing technical infrastructures [9]. Key metadata elements and functional groups required to describe health datasets comprehensively were identified and documented.

**Phase 2: Stakeholder engagement and Use-Case definition**
Extensive stakeholder engagement was conducted following an Agile methodology, including workshops, structured surveys, and iterative feedback loops [10]. This incremental development process involved close collaboration with representatives from HDABs, dataset catalogue operators, researchers, data holders, data users, health experts and policymakers. Stakeholders jointly defined critical use cases, ensuring that the new HealthDCAT-AP properties addressed concrete, real-world scenarios for dataset discovery and reuse.

**Phase 3: Initial specification and controlled vocabularies development**
Building on clearly defined use cases and stakeholder-validated requirements, the HealthDCAT-AP extension specification was drafted using the DCAT-AP template [2] as a foundation. This involved specifying new classes and properties, controlled vocabularies, usage notes and cardinalities. SHACL-based validation rules were created to enforce metadata consistency and quality. The draft HealthDCAT-AP specification[24] was published openly on Github documented using ReSpec[25] offering as a Single Source Of Trust and facilitating transparency and collaborative refinement. Feedback was actively collected through GitHub and systematically addressed throughout the iterative development process.

---

[23] SEMIC | Interoperable Europe Portal
[24] Draft HealthDCAT-AP Specification
[25] ReSpec Documentation

**Phase 4: Validation through sandbox implementation**

A "TO-BE" sandbox dataset catalogue and a dedicated HealthDCAT-AP metadata editor[26] with integrated validation capabilities, enabled stakeholders to test and refine the specification using real-world datasets. A sandbox environment was organised through dedicated national catalogues, each implementing and testing the HealthDCAT-AP specification. The validation process was further supported by mapping exercises, aligning metadata from existing dataset catalogues to the HealthDCAT-AP model. In parallel to the sandbox validation, the European Commission developed a reference technical stack for implementing HealthDCAT-AP compliant national dataset catalogues. This stack includes key technical features aligned with the HealthDCAT-AP specification - such as advanced filtering capabilities that allow users to search datasets by criteria like age range, disease, or legal basis - demonstrating the practical application of the extended metadata elements and supporting consistent implementation across Member States [11, 12]. This phase was essential for testing the practical applicability and interoperability of the new metadata terms and controlled vocabularies. Feedback from real-world usage was systematically documented, enabling iterative improvements and refinements.

**Phase 5: Evaluation, implementation challenges, and adoption pathways**

Following sandbox validation, the HealthDCAT-AP specification was further evaluated and refined through the TEHDAS2 Joint Action. A series of technical workshops with experts from multiple Member States enabled detailed discussion of the specification, complemented by surveys and follow-up exchanges. A pan-European public consultation extended this evaluation beyond the project partners, collecting feedback on both the structure of the specification, down to the property level, and on broader implementation challenges. Additional input came from data holders actively working to create HealthDCAT-AP-compliant metadata records, helping to identify practical challenges and refine the supporting materials. This included the development of a comprehensive guideline to support data holders in applying the model effectively. The process also highlighted wider technical and organisational considerations necessary for successful implementation within the EHDS, even if not directly related to the specification itself. This work is intended to inform the Commission's implementing acts under Article 77(4), formally embedding HealthDCAT-AP into the EU regulatory framework. These implementing acts will be adopted following the examination procedure referred to in Article 98 of the EHDS Regulation. A governance model will be established, transferring responsibility for the ongoing maintenance of HealthDCAT-AP to the European Commission, which will ensure its continued alignment with evolving EU regulations, policy priorities, and technical standards.

This structured roadmap ensured that every stage of HealthDCAT-AP's development was transparently documented, iteratively validated against concrete implementation experiences, and directly informed by the community of stakeholders it was designed to serve.

## 4.2.    Involving relevant stakeholders for Use-Case and requirements definition

Stakeholder engagement was central to the successful definition of use cases and elicitation of functional requirements for HealthDCAT-AP. Structured stakeholder involvement included the following key activities:

- **Technical Working Group:**
    - A dedicated Technical Working Group (TWG) of approximately 80 participants, including national HDAB representatives, data holders, researchers, dataset publishers, policymakers, and technical experts, provided continuous guidance. The TWG convened twelve times in a period of six months, systematically addressing critical topics such as dataset identifiers, health terminologies, data sensitivity classifications, population

---

[26] HealthDCAT-AP | Health Information Portal

coverage metrics, quality annotations, analytics capabilities, and access regulatory procedures.

- Each TWG session utilised EU-Survey forms that collected structured feedback from stakeholders, enabling iterative review and refinement of the proposed metadata model [10].

- **Persona-based requirement elicitation:**
  - Stakeholders collaboratively developed concrete, persona-based scenarios ("As a researcher, I want...") to ensure that the metadata model directly addressed real-world functional requirements.
  - These scenarios validated essential metadata groups, including data discovery and annotation, data access, provenance, ownership identification, temporal/spatial/population coverage, data analytics, data quality, and variable-level descriptions.

- **Multi-layered consultation and validation:**
  - The draft specification was publicly available on GitHub, allowing open tracking of issues, proposals for amendments, and collaborative refinement.
  - A practical sandbox environment enabled non-technical stakeholders to apply and evaluate HealthDCAT-AP metadata elements directly on their datasets, providing immediate feedback regarding usability and completeness.
  - Structured interviews and consultations with SEMIC27 and DCAT-AP experts ensured alignment with broader interoperability frameworks and best practices, particularly concerning FAIR data principles and regulatory compliance.

- **Iterative validation tools:**
  - An online validation service systematically compared metadata records between "AS-IS" DCAT-AP and "TO-BE" HealthDCAT-AP states, quantitatively highlighting missing mandatory elements and metadata quality improvements. This approach directly translated stakeholder feedback into measurable quality enhancements, fostering practical improvements.

This stakeholder-driven approach ensured that HealthDCAT-AP not only met technical and regulatory requirements but also directly addressed real-world user needs and expectations, significantly enhancing its practical impact and adoption prospects.

## 5. Definition of the Requirements

The specification of HealthDCAT-AP relied on a rigorous, structured definition of metadata requirements, derived from a comprehensive combination of functional grouping, real-world use cases, adherence to FAIR data principles, explicit EU policy mandates, and technical considerations. This thorough approach ensured the metadata extension would comprehensively meet user, regulatory, and operational needs without introducing unnecessary complexity. These key design principles provide a replicable blueprint for future domain-specific extensions.

### 5.1.  Identification of metadata elements by functional groups

HealthDCAT-AP systematically categorised metadata elements into clearly defined functional groups, aligning directly with practical user needs and regulatory requirements. These functional groups included:

- **Data discovery:** Metadata enhancing dataset discoverability, including thematic classifications, multilingual titles, keywords, and structured descriptions.
- **Data annotation:** Explicit linkage to standardised terminologies and coding systems (e.g., ICD-10, SNOMED CT).

---

[27] SEMIC is responsible for the maintenance and publication of DCAT-AP and other Application Profiles such as DCAT-AP for High-Value Datasets (HVD) or GeoDCAT-AP.

- **Data access:** Metadata defining access rights, conditions, access regulatory procedures including explicit references to HDABs, and data sensitivity levels (open, protected, sensitive), alongside explicit retention periods.
- **Data provenance and ownership:** Metadata capturing dataset origin, creation processes, and stewardship responsibilities.
- **Temporal and spatial coverage:** Detailed metadata regarding the periods and geographic areas described by datasets, supporting precise filtering and selection by users.
- **Population coverage:** Explicit descriptions of population characteristics (age ranges, demographic, disease-specific cohorts) to facilitate clinical and epidemiological research queries.
- **Data analytics:** Availability of analytics-ready data and suitability for secondary uses.
- **Data quality:** Metadata elements reflecting structured quality annotations and compliance indicators, directly supporting EHDS regulatory quality and utility labels (Article 78).
- **Variable-level description:** Structured metadata describing dataset structure, sample datasets for preliminary exploration.
- **Data categorisation:** Explicit domain, health-specific themes according to Article 51, and legal categorisation.

Grouping metadata elements by these clearly defined functional groups ensured comprehensive, coherent, and practical coverage of all EHDS-mandated metadata requirements (See Annexes – Table2 | Metadata elements grouped by functional groups).

## 5.2. Review of Use Cases

The definition of HealthDCAT-AP's metadata elements was driven by detailed persona-based use cases articulated as "As a <role> I want … so that …" statements. Through targeted surveys and workshops involving stakeholders, practical user stories were collected and validated - for example, researchers needing detailed population coverage information for cohort selection (*healthdcatap:populationCoverage*), or policymakers seeking datasets tagged explicitly for policy analysis (*dcatap:applicableLegislation*), or the need for analytics metrics to support feasibility research studies (*healthdcatap:analytics*). Each metadata element introduced into HealthDCAT-AP directly responds to validated user stories, ensuring real-world relevance and practical utility. In addition to stakeholder engagement, the TWG reviewed several existing dataset catalogues as part of the elicitation of use cases related to data in scope of Article 51 of the EHDS Regulation. These included:

- The Health Information Portal on Population Health[28]
- The Health Data Research UK (HDR UK) Catalogue[29]
- The BBMRI-ERIC Biobank Catalogue[30]
- The EJP RD Virtual Platform on Rare Diseases[31]
- And others relevant to research and public health such as the Norwegian health data portal HelseData[32].

This review ensured alignment with established practices and helped identify gaps that HealthDCAT-AP could address.

Examples of Use Cases include:

---

[28] https://healthinformationportal.eu
[29] https://www.hdruk.ac.uk/
[30] https://directory.bbmri-eric.eu
[31] https://vp.ejprarediseases.org/
[32] https://helsedata.no/en/

- **Researcher:** discover cohorts with ≥10 000 cases of type-2 diabetes across ≥3 Member States for an AI-driven outcome-prediction study.
- **HDAB catalogue manager:** monitor datasets whose retention period will expire within six months, in order to exclude them from eligibility for access requests.

## 5.3.    Requirements derived from FAIR Data Principles

HealthDCAT-AP explicitly incorporates the FAIR data principles. This alignment ensures that datasets described with HealthDCAT-AP metadata are inherently FAIR, facilitating their discoverability (through standardised vocabularies), accessibility (via clear data access conditions), interoperability (using persistent identifiers and machine-actionable controlled vocabularies), and reusability (clearly specifying provenance, usage purposes, and quality annotations). The explicit adherence to FAIR principles directly guided the mandatory inclusion of certain metadata elements:

- **Findable:** Unique, persistent, dereferenceable dereferenceable identifiers (*dct:identifier*), multilingual metadata fields, standardised thematic tags associated with mandatory controlled vocabularies, and machine-actionable metadata.
- **Accessible:** Clearly defined access conditions (*dct:accessRights*), explicit retention periods (*healthdcatap:retentionPeriod*), and secure, standardised data distribution links.
- **Interoperable:** Use of RDF-based structured metadata, multilingual controlled vocabularies (Publications Office, Wikidata, EuroVoc, etc.), and explicit coding-system linkages for semantic clarity (ICD-10, SNOMED CT).
- **Reusable:** Detailed provenance metadata (*dct:provenance*, *prov:wasGeneratedBy*), clear licensing conditions (*dct:license*), structured quality annotations (*dqv:hasQualityAnnotation*), and mandatory data dictionaries and sample datasets for preliminary exploration (*adms:sample*) for non-public datasets.

## 5.4.    Requirements derived from EU Policies

HealthDCAT-AP explicitly addresses and aligns with metadata obligations derived from multiple EU regulatory frameworks. Foremost, it is purpose-built to support the EHDS, explicitly addressing and aligning with the metadata obligations set out in the EHDS Regulation. In addition, it addresses the metadata requirements stemming from horizontal EU data policies:

- **EHDS Regulation** [1]**:** The EHDS Regulation provided explicit regulatory mandates guiding HealthDCAT-AP's requirements, notably Article 77 (minimum elements health data holders are to provide for datasets), Article 78 (quality and utility labels), Article 60 (dataset description annual review), Article 51 (health-specific categories) and Article 80 (high-impact dataset specifications). Additionally Article 77(3), alignment with broader EU data governance frameworks - such as the Data Governance Act (DGA) Article 8 (Single Information Points) - was explicitly maintained.
- **Data Governance Act (DGA)** [3]**:** Alignment with Single Information Points, explicit dataset publisher, and clearly defined access procedures, rights and reuse conditions.
- **High-Value Dataset (HVD) Implementing Regulation** [8]**: Ensuring** datasets explicitly flagged through mandatory European Legislation Identifier (ELI) tagging, supporting legal compliance and discoverability.
- **General Data Protection Regulation (GDPR)** [4]**:** Metadata explicitly representing sensitive data safeguards, the legal basis for data collection, clearly defined retention periods, and the specific purpose for which the data was collected.
- **Data Act** [13]**:** Alignment with data transparency requirements by ensuring that data structures, formats, vocabularies, and classification schemes are described in a publicly available and consistent manner.

These explicit regulatory requirements directly informed the definition of new mandatory and recommended metadata properties in HealthDCAT-AP.

## 5.5. Requirements derived from technical considerations

Finally, the TWG assessed the technical features needed to support health dataset catalogues specifically and next-generation catalogues. These requirements informed the selection and design of metadata elements and controlled vocabularies and structural patterns in HealthDCAT-AP, ensuring user-centric, efficient, and interoperable dataset catalogues.

Key areas include:

- **Search and discovery technologies:** HealthDCAT-AP supports a wide range of search capabilities to meet diverse user needs:
- **Keyword and full-text search:** Enabled by metadata fields of range Literal such as title, description, provenance, population coverage and purpose.
- **Faceted search:** Allowing to filter by dimensions like EHDS data categories, health themes, age range, and temporal or spatial coverage, etc.
- **Semantic search and SPARQL queries:** Leveraging semantic relationships between concepts using standard vocabularies and persistent URIs, supporting advanced users with complex query needs.
- **Natural Language Processing (NLP) and Artificial Intelligence (AI):** Enabling intelligent query interpretation and relevance ranking through machine learning techniques.
- **Geospatial search:** Facilitated through precise geographic metadata, allowing datasets to be located and filtered by region or coordinates.

- **Knowledge graph interoperability and federation:** HealthDCAT-AP is designed for use in RDF-based knowledge graphs with persistent, dereferenceable URIs, enabling cross-catalogue linking, federated querying, and duplicate detection across Member States.

- **Multilingual metadata:** Language-tagged literals and multilingual thematic classifications enabling compliance with EHDS Article 77 (2).

# 6. Case Study: HealthDCAT-AP

The HealthDCAT-AP extension was specifically designed to address the detailed health-domain metadata requirements identified in alignment with the EHDS Regulation [12]. This chapter presents an overview of the main additions introduced by HealthDCAT-AP, including new metadata properties, controlled vocabularies, refined usage notes and cardinalities, and illustrative examples.

## 6.1. New properties

HealthDCAT-AP fully reuses all core classes and properties defined in DCAT-AP, while extending the model with new metadata elements specifically tailored to health data. These health-specific extensions are primarily added to the class dcat:Dataset, complementing the generic DCAT-AP vocabulary to meet the key requirements identified in the context of the EHDS. A total of 20 new properties have been introduced in HealthDCAT-AP that were not previously defined by DCAT-AP requirements. These include:

- For the class *dcat:Dataset*:
*dct:alternative, healthdcatap:analytics, healthdcatap:codeValues,*
*healthdcatap:hasCodingSystem, healthdcatap:healthCategory, healthdcatap:hdab,*
*healthdcatap:healthTheme,dpv:hasLegalBasis, healthdcatap:maxTypicalAge,*
*healthdcatap:minTypicalAge, healthdcatap:numberOfRecords,*
*healthdcatap:numberOfUniqueIndividuals, dpv:hasPersonalData,*
*healthdcatap:populationCoverage, dpv:hasPurpose, dqv:hasQualityAnnotation,*
*healthdcatap:retentionPeriod.*

- For the class *foaf:Agent*:
*healthdcatap:publisherNote, healthdcatap:publisherType, healthdcatap:trustedDataHolder.*

Below are illustrative examples of how these properties address EHDS-specific needs:

- **Identification and governance:**
  - *healthdcatap:hdab*: References the legally competent HDAB.
  - *healthdcatap:publisherType*, *healthdcatap:publisherNote*: Provide clarity on the role and nature of the dataset publisher.

- **Categorisation:**
  - healthdcatap:healthCategory: Supports categorization of health data in line with Article 51 of the EHDS Regulation.
  - healthdcatap:healthTheme: Facilitates multilingual tagging and enables faceted search.

- **Population and cohort metrics:**
  - *healthdcatap:minTypicalAge*, *healthdcatap:maxTypicalAge*: Defines the typical age range covered by the dataset.
  - *healthdcatap:populationCoverage*: Describes the demographic or cohort characteristics.
  - *healthdcatap:numberOfRecords*, *healthdcatap:numberOfUniqueIndividuals*: Provide basic analytical metadata to support feasibility assessments.

- **Analytical and technical context:**
  - *healthdcatap:analytics*: Metadata of class "Distribution", provides analytical metrics supporting dataset discoverability.
  - *adms:sample*: Enables linking to synthetic or mock-up sample datasets and/or to data dictionaries expressed using the CSVW RDF vocabulary.
  - *healthdcatap:hasCodingSystem*: Specifies the coding systems and terminologies applied to variables (e.g. ICD-10-CM, SNOMED CT).
  - *healthdcatap:hasCodeValues*: Enables annotation of datasets using standard terminologies, thesauri, and classification lists, providing a semantic layer that supports automated search and reasoning in data catalogues.

- **Compliance and lifecycle management:**
  - *dpv:hasPurpose*: Describes the intended primary purposes of data collection and use.
  - *dqv:hasQualityAnnotation*: Supports structured annotation of data quality and compliance attributes.
  - *healthdcatap:retentionPeriod*: States the dataset's retention policy, if applicable.
  - *dpv:hasLegalBasis*: Refers to the legal basis governing the processing of the health data.

All these additions enrich the dataset description by providing the additional context required by the EHDS Regulation and the specific needs of health data, while remaining fully backward compatible with generic DCAT-AP implementations and harvesters that only support the DCAT-AP vocabulary.

## 6.2. New Controlled Vocabularies

To ensure consistent and interoperable metadata descriptions across national dataset catalogues, HealthDCAT-AP introduces mandatory use of new controlled vocabularies. These vocabularies promote semantic consistency, support multilingual metadata, and enable automated validation. Each controlled vocabulary is aligned with international standards and maintained in EU multilingual Simple Knowledge Organization System (SKOS) format:

- **healthCategory:** A SKOS vocabulary aligned with Article 51 of the EHDS Regulation, covering categories such as Electronic Health Records and Genomic Data.
- **healthTheme:** References to health concepts using persistent, multilingual identifiers from established Wikidata knowledge graphs platform.
- **hdab:** An authoritative registry of Health Data Access Bodies, maintained by the European Commission.
- **publisherType:** A dedicated taxonomy for health-related organisation types, including hospitals, research infrastructures, and biobanks.

In addition, HealthDCAT-AP mandates the use of the following controlled vocabulary:

- **accessRights:** Standardised EU Rights Statements (open, restricted, non-public) as published by the Publications Office of the EU, ensuring harmonised access rights in line with EHDS requirements.

Furthermore, the DCAT properties prov:qualifiedAttribution (used to describe agents with specific responsibilities for the resource) and dcat:qualifiedRelation (used to describe relationships with other resources) would benefit from the introduction of dedicated controlled vocabularies tailored to the health domain.

## 6.3.    Properties under headings mandatory, recommended, optional & usage notes

HealthDCAT-AP introduces refined groups of mandatory, recommended, optional properties, with defined cardinalities, specifying which metadata elements under defined conditions. These refinements aim to ensure rich, machine-actionable metadata while minimizing unnecessary burden on data providers. HealthDCAT-AP distinguishes between **open**, **protected**, and **sensitive** datasets based on the value of *dct:accessRights*, and introduces three corresponding conditional minima.

**Baseline mandatory elements (All Datasets)**
DCAT-AP mandates only *dct:title* and *dct:description* as required properties. In HealthDCAT-AP, the following properties for the class *dcat:Dataset* are mandatory for all datasets - regardless of their sensitivity level - to comply with EHDS elicited requirements:

- *dct:identifier*
- *dct:title*
- *dct:description*
- *dcat:theme*
- *dct:accessRights*
- *dcat:distribution*
- *healthdcatap:hdab*
- *dcatap:applicableLegislation*
- *healthdcatap:healthCategory*

**Conditional requirements based on access level (Protected and Sensitive Datasets)**

- *dct:publisher* is mandatory for all datasets that are not open.
- **Sensitive Datasets** (e.g., involving personal data):
  Several additional properties become mandatory or recommended to enhance discoverability and enable feasibility assessments. For instance:
- At least one synthetic or mock-up sample or data dictionary distribution must be provided using *adms:sample*.
- Publishers are encouraged to expose analytics endpoints (e.g., API-based usage statistics or sample queries) to support early-stage feasibility checks.

While DCAT-AP provides general usage notes for its properties, HealthDCAT-AP refines these notes to better align with the specific needs of health data and the requirements of the EHDS Regulation. Usage notes in HealthDCAT-AP offer more precise guidance to ensure a better consistency. For example, the usage note for dcat:theme not only confirms that multiple themes may be assigned to a dataset but also mandates the use of the EU's controlled vocabulary for Data Themes - requiring the inclusion of the value NAL:data-theme[33] "HEAL" for health-related datasets. Similarly, the usage note for dct:provenance specifies that this property should describe how the data was

---

[33] EU vocabularies : data-theme

collected, including methodologies, tools, and protocols, thus promoting transparency and trust in the dataset's origin.

## 6.4. Key Examples

Illustrative examples demonstrating HealthDCAT-AP metadata properties in practical dataset descriptions are provided in the Table 3 of annexes (Examples of HealthDCAT-AP metadata properties of the class Dataset)

# 7. Evaluation & lessons for other domains

The HealthDCAT-AP case study provides important insights and transferable lessons for future domain-specific DCAT-AP extensions. These insights have clear, transferable lessons for other domains.

## 7.1. Evaluation of HealthDCAT-AP

HealthDCAT-AP was carefully designed and thoroughly validated within a structured pilot framework supported by a multi-country sandbox environment. This evaluation involved stakeholders actively transforming real-world dataset metadata records from existing templates ("AS-IS") to the new HealthDCAT-AP profile ("TO-BE"), publishing records to the practical sandbox catalogue, and testing metadata harvesting into a prototype EU Health Dataset Catalogue. This iterative testing approach confirmed several significant benefits:

- **Enhanced discoverability and interoperability:** The introduction of precise controlled vocabularies and structured metadata significantly improved dataset searches, enabling users to filter efficiently.

- **Compliance with regulatory frameworks:** HealthDCAT-AP effectively incorporates the requirements of the EHDS and other cross-sectoral data regulations directly into metadata descriptions - most notably through conditional obligations based on the dataset's access level [12].

- **Effectiveness of editing and validation tools:** The availability of a dedicated HealthDCAT-AP editor significantly improved the quality and consistency of metadata authoring. Combined with integrated online validation tools, it enabled users to systematically detect and resolve metadata quality issues like missing mandatory elements. This iterative process led to measurable improvements in metadata completeness and conformance across successive testing cycles.

- **Sensitive data handling:** As an innovation within the DCAT-AP framework, preliminary business analysis confirmed the value of using synthetic dataset samples (adms:sample) to describe data structures, formats, vocabularies, and classification schemes. When combined with structured analytics metadata (healthdcatap:analytics), this approach enabled meaningful exploration and discovery without compromising data privacy. It represents a promising, reusable best practice for managing sensitive or protected datasets, pertinent to many domains and in line with the DGA's focus on lowering barriers for protected data reuse

- **Controlled vocabulary governance:** Stakeholders positively validated controlled vocabularies derived from EHDS Article 51, Wikidata health concepts, and health-specific organisation types. However, the pilot highlighted the critical necessity for a clear governance model, including long-term curation and version management strategies [14]

- **AI-optimized metadata:** Pilot implementations confirmed the suitability of HealthDCAT-AP metadata properties for supporting semantic and generative AI-based discovery tools, although further benchmarking was suggested.

- **Ease of implementation:** The minimalist extension approach will facilitate rapid adoption. Stakeholders praised the targeted enhancements that minimized the effort needed to upgrade existing DCAT-AP catalogues.

- **Metadata quality assessment:** HealthDCAT-AP was evaluated against the EU Data Portal's metadata quality framework, with a specific focus on aligning its metadata elements to the quality and utility labels defined under Article 78 of the EHDS Regulation. This alignment led to significant improvements in both metadata completeness and semantic richness.

However, the HealthDCAT-AP experience also identified limitations and challenges:

- **Data holder diversity:** Publishers with less technical maturity initially faced challenges adopting RDF-based metadata management.

- **Ongoing governance:** Continuous stakeholder engagement and active management of controlled vocabularies are essential, requiring sustained coordination.

- **Scalability considerations:** Large-scale catalogue implementations need robust technical infrastructure, particularly for validation and querying.

## 7.2. Transferable Lessons

The experience from developing and validating HealthDCAT-AP resulted in several practical and transferable lessons:

**Lesson 1: Minimal and targeted extensions**
HealthDCAT-AP demonstrated the value of carefully selecting only necessary metadata elements that directly address validated user and regulatory requirements. Minimal extensions reduce complexity, accelerate implementation, and increase user acceptance.

**Lesson 2: Namespace isolation and semantic stability**
Clearly separating new properties into dedicated namespaces ensures stability and avoids unintended semantic clashes. This preserves backward compatibility, enabling seamless integration into existing DCAT-AP-based catalogues.

**Lesson 3: Early stakeholder engagement and iterative design**
Involving relevant stakeholders through structured surveys, workshops, and iterative feedback loops was critical for ensuring HealthDCAT-AP's practical utility. Stakeholder-driven use-case analysis ensures real-world relevance, accelerating adoption.

**Lesson 4: Leveraging open and established controlled vocabularies**
Reusing existing international controlled vocabularies (e.g., ICD-10, SNOMED CT, EuroVoc) enhances semantic interoperability. Maintaining multilingual SKOS vocabularies further improves dataset discoverability across linguistic barriers.

**Lesson 5: SHACL-based validation tooling**
Comprehensive validation with SHACL ensures consistent metadata quality and rapid identification of non-conformities. Providing ready-to-use validation tools significantly simplified implementation and compliance monitoring across catalogues.

**Lesson 6: Structured yet lightweight governance**
Establishing a transparent, inclusive, and lightweight governance structure (e.g., editorial boards, open GitHub repositories, public issue trackers) facilitated rapid consensus, transparency, and sustainable maintenance of the extension.

**Lesson 7: Practical tooling for adoption and compliance**
To meet their EHDS obligations, data holders require more than just a specification - they need practical, easy-to-use tools. The provision of a dedicated metadata editor and a lightweight catalogue solution proved essential for enabling metadata creation, validation, and publication with minimal

technical overhead. These tools lower the barrier to entry, especially for smaller organisations and decentralised infrastructures, and ensure consistent implementation of HealthDCAT-AP across diverse institutional contexts. Tooling is not an optional extra - it is a critical enabler of adoption and compliance.

### 7.3. Implications for other domains

Other sectors planning DCAT-AP extensions can directly benefit from these lessons. A minimalistic, targeted, and stakeholder-driven approach ensures rapid, effective, and sustainable metadata enhancements, closely aligned with domain-specific regulatory and technical requirements. Such carefully scoped extensions reduce implementation overhead while significantly improving metadata discoverability, usability, and interoperability at scale.

In summary, HealthDCAT-AP provides a robust blueprint for domain-specific DCAT-AP extensions, demonstrating that carefully defined minimal extensions, guided by clear design principles and supported by practical validation tools, can deliver substantial, immediate benefits in terms of interoperability, compliance, and practical usability across European data spaces.

## 8. Conclusions

The HealthDCAT-AP case study demonstrates the effectiveness of a structured and stakeholder-driven approach for extending DCAT-AP for the EHDS. By introducing only essential metadata elements, HealthDCAT-AP addresses the specific legal, ethical, and operational requirements set out by the EHDS Regulation, while maintaining full compatibility with existing European metadata infrastructures.

The success of HealthDCAT-AP lies not only on its technical design but also in the inclusive process by which it was developed. Early and sustained stakeholder engagement, iterative validation in real-world sandbox environments, the development of a a HealthDCAT-AP metadata editor and a SHACL-based automated validation tooling (machine-readable rules that check RDF for compliance) have ensured that the profile remains both practical and compliant. Its release as a single source of truth, supported by open governance and a ready-to-use validation toolkit, facilitates adoption by Health Data Access Bodies and data holders across the EU.

HealthDCAT-AP thus provides a replicable blueprint for future domain-specific DCAT-AP extensions. Its design principles - minimalism, semantic clarity, reuse of established vocabularies, and emphasis on interoperability - can guide similar efforts in other sectors, from energy and environment to finance and education. In doing so, it contributes to the broader vision of interoperable, federated European Data Spaces, supporting responsible data reuse, innovation, and policy development across borders.

## Acknowledgements

We are grateful to Pavlina Fragkou, Bert Van Nuffelen, and Makx Dekkers for their steadfast contributions, to Andrea Perego for his insightful early-stage advice, and to the wider Semantic Interoperability Community (SEMIC) for their continuous support.

## Declaration on Generative AI

During the preparation of this work, the author(s) used CHAT-GPT-4 in order to: Grammar and spelling check. After using these tool(s)/service(s), the author(s) reviewed and edited the content as needed and take(s) full responsibility for the publication's content.

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

## Annexes:

**Table 2**
Metadata elements grouped by functional groups

| Functional group | Purpose (excerpt) | DCAT example properties: Core and extended * | Key remarks |
|---|---|---|---|
| Data Discovery | Make the dataset findable by theme, keyword or free text | *dct:title, dct:description, dcat:keyword, healthdcatap:healthTheme* | Must support multilingual values and automated keyword tagging |
| Data Annotation | Make the metadata machine actionable | *dct:conformsTo, healthdcatap:hasCodingSystem, healthdcatap:hasCodeValues* | Wikidata concept URIs MUST be used |
| Data Access | Tell users how and under what conditions they can retrieve data | *dcat:distribution, dct:accessRights, odrl:hasPolicy, healthdcatap:retentionPeriod* | Distinguish open / protected / sensitive data access flows |
| Data Provenance | Trace origin and lifecycle | *dct:provenance, prov:wasGeneratedBy, dct:source, dct:creator* | Essential for EHDS Article 78 quality labels |
| Data Ownership | Identify controller & steward | *dct:publisher, healthdcatap:hdab* | Must link to HDAB register |
| Temporal Coverage | Period the data describe | *dct:temporal, dct:issued, dct:modified* | Needed for coverage part of quality label |
| Spatial Coverage | Geography described | *dct:spatial* | Support bounding boxes & coded regions |
| Population Coverage | Characteristics of cohort | *healthdcatap:populationCoverage, healthdcatap:minTypicalAge, healthdcatap:maxTypicalAge* | Drives faceted clinical search |
| Data Analytics (metrics) | Summarise size & structure | *healthdcatap:numberOfRecords, healthdcatap:numberofUniqueIndividu* | Helps feasibility checks |

| | | | |
|---|---|---|---|
| | | *als, healthdcatap:analytics* | |
| Data Quality | Express quality & utility | *dqv:hasQualityAnnotation,* data-quality certificate | Link to EHDS Article 78 label |
| Variables | Describe data schema | *adms:sample,* CSV-on-the-Web terms | Mandate data dictionary for non-public data |
| Data Categorisation | Classify by legal/domain | *dct:theme, healthdcatap:healthCategory, dcatap:applicableLegislation* | Enables cross-domain harvesting |

* Extended properties are those proposed in the draft profile. The namespace for HealthDCAT-AP is associated with the prefix "healthdcatap".

**Table 3**
Examples of HealthDCAT-AP metadata properties of the class Dataset

| DCAT property | Example |
|---|---|
| Identifier | *dct:identifier "https://fair.healthdata.be/dataset/d43a158e-7d13-4660-bbc3-9d3f8d55015"^^<http://www.w3.org/2001/XMLSchema#anyURI>;* |
| Theme | *dcat:theme <http://publications.europa.eu/resource/authority/data-theme/HEAL>;* |
| Coding System | *healthdcatap:hasCodingSystem <https://www.wikidata.org/entity/P1690>;* |
| Health Theme | *healthdcatap:healthTheme <https://www.wikidata.org/entity/Q58624061>,<https://www.wikidata.org/entity/Q7907952>;* |
| Personal Data** | *dpv:hasPersonalData dpv-pd:Gender, dpv-pd:Age, dpv-pd:Location, dpv-pd:Nationality, dpv-pd:Education, dpv-pd:HealthRecord;* |
| Sample | *adms:sample [ a dcat:Distribution ; dct:description "Proxy data generating for the EHDS2 Pilot project Sciensano Use Case"@en; dcat:downloadURL <https://github.com/CAVDgit/EHDS2_UC_Sciensano/blob/main/use_case_1_synthetic_data_10K_individuals.csv>; dcat:mediaType <http://www.iana.org/assignments/media-types/text/tab-separated-values> ];* |
| Population Coverage* | *healthdcatap:populationCoverage* |

| | |
|---|---|
| | *"Adults (age 18-65) from EU Member States diagnosed with Type 2 Diabetes, 2015-2022."@en* |
| Purpose* | *dpv:hasPurpose [ a dpv:Purpose;*
*dct:description "The primary objective of Sciensano's LINK-VACC project is to monitor COVID-19 vaccines post-authorization and evaluate the public health value of prioritising vaccination for people with comorbidities. This involves assessing the vaccines' effectiveness and safety in the broader population context, beyond the limited scope of clinical trials, and determining future vaccination policies in public health emergencies such as epidemics or pandemics"@en ];* |

* New full-text properties included alongside the property Description (*dct:description*).

** The Data Privacy Vocabulary (DPV) enables expressing machine-readable metadata about the use and processing of (personal or otherwise) data and technologies. The Personal Data (PD) extension provides additional concepts to represent different types and categories of personal data for use with the Data Privacy Vocabulary (DPV) Specification.