# OpenReview forum: "Designing DCAT-AP extensions for common European data spaces: The EHDS HealthDCAT-AP Case Study"
_SEMANTiCS.cc/2025/Workshop/NXDG — NXDG 2025_

### Official Review · ~Chang_Sun1 · 2025-07-16
**A nice full paper on DCAT-AP extension for EHDS**

**Rating:** 7
**Confidence:** 3

**Review:**

This paper presents the development and validation of HealthDCAT-AP, a health-specific extension of the widely adopted DCAT-AP metadata standard, tailored for EHDS. The result shows a domain-specific metadata profile to address EHDS regulatory obligations and facilitate interoperability across EU health data catalogues.

The paper is aligned with the scope of NXDG workshop by proposing a concrete example of data governance. The paper provides sufficient background and details of EU health data regulations, DCAT-AP which makes it easy for readers to understand the challenges and research gaps. The multi-phase roadmap (from requirements elicitation to stakeholder consultation to validation) is exemplary and could serve as a blueprint for other domain-specific extensions. The requirements from different aspects are clearly explained and connected with the work. The proposed extension preserves compatibility with DCAT-AP and aligns with key principles such as namespace isolation, controlled vocabularies, and minimal, justified extensions. The important part is to clearly map to EHDS legal requirements (e.g. Articles 60, 77–80) and other EU frameworks (GDPR, DGA, Data Act) makes this work timely and sound.

Although the conceptual architecture and metadata properties are well described, the manuscript is lack of a more detailed breakdown of technical implementation challenges (e.g. SHACL validation complexity, vocabulary governance issues).
The manuscript mentions HealthDCAT-AP metadata supports AI-based tools, but provides no concrete implementation or further discussion in this regard.

Would it be possible to compare the proposed work with other DCAT-AP profiles  or to alternative metadata standards used in health (e.g. HL7 FHIR)?? This could strenghten the evaluation of the work.

In general, I think the work is decent and worth to present in the workshop.

---

### Official Review · ~Rob_Brennan1 · 2025-07-23
**An important contribution to our domain**

**Rating:** 9
**Confidence:** 5

**Review:**

This paper discusses the background and development of Health DCAT-AP, the new data catalogue metatdata specification to support the EHDS. It provides an overview of the processes used, requirements gathered and illustrates the use of the new profile.

**This work is an important contribution and it is great for the workshop to have the opportunity to publish it.**

It is a highly readable paper. My only caveats are
1. Many items of the no doubt long gestation have only been covered in summary form due to space limitations. I am sure a book could be written on the full process, arguments and counter-arguments of the many design choices made.
2. I have some small comments on some of the technical choices made, many of these will no doubt have been already hashed through as part of the design process. However it shows that the current format does have leave out some aspects of the design due to space considerations.
3. There a couple of places where I think that minor over-claims are made by the authors but this is partially due to the ongoing nature of the work
a) An appeal to minimalism is made many times - but 20 new fields have been added. This makes Health DCAT-AP very cumbersome. There is at least 1 field that could be a sub-property.
b) Not all the controlled-vocabularies are full available publicly yet, perhaps these exist internally.
c) Implementation notes are claimed as more complete than DCAT-AP, that is true but there are also many new areas of ambiguity that have been opened up due to the increased scope. No doubt these will be fixed over time (and there is a balance to be had with enforcing too much of a burden on people seeking to deploy it).

Well done! I really like the sub-profiles and inclusion of data protection metadata in the catalogue. In practice I think this is key to decision-making about the (re-)usability of data in a modern context.

=Detailed comments

p4 "Consequently, DCAT-AP does not directly replace or manage operational data systems;"
This is slightly confusing. DCAT-AP is a specification. It cannot replace or manage operational systems but an IT system/product conformant to DCAT-AP could do those things. Please clarify.

p12, Comments on some of these:
Why is healthTheme not a sub-property of dcat:theme to show they are semantically related? This seems to be evidence of closed world rather than open world modelling (when open world is the default for Sem Web).

healthTheme: It does not seem appropriate to me to make use of wikiData values mandatory for fields. It is fine for it to be allowed or recommended, but preclusion of other, possibly more authoritative sources (eg national authorities, standards bodies) could lead to issues in the future where things are not in WikiData and are not most properly defined there.
Quoting from the specification "Wikidata concept URIs MUST be used for the following properties coding system, code values, conform to, health theme. "

dpv:hasPurpose, dpv:hasLegalBasis - the examples provided in the specification imply that these can essentially be strings rather than enumerated types as per DPV. This reduces machine-readability, which is one of your goals.

publisherType: - has this vocabulary been defined yet? The draft says it is TBD.

p13, Mandatory elements
Having applicable legislation mandatory for all datasets seems unusual when we go down to the granularity of national datasets or even ones collected within a single organisation. Not all datasets have legislation associated with them, just like not all datasets are personal. This makes sense for EU level things managed by the Commission but HealthDCAT-AP is going to have much broader use.

p13 "Usage notes in HealthDCAT-AP offer more precise guidance to ensure a better consistency."
In practical terms these could still be improved. There are many cases where there are multiple options mentioned for filling in fields and it is not obvious what is the default or preferred eg Frequency.
In general the lack of specification will lead to interoperability challenges in the future. Recall John Postel's motto that you should be conservative in what you send but liberal in what you receive. If there are a lack of concrete guidelines there will be less interoperability. This does not have to be at the expense of exclusivity, but just having a default or recommendation.

---

### Decision · Program_Chairs · 2025-07-25

Accept